# Benefits of Maternal Choline Supplementation on Aged Basal Forebrain Cholinergic Neurons (BFCNs) in a Mouse Model of Down Syndrome and Alzheimer’s Disease

**DOI:** 10.3390/biom15081131

**Published:** 2025-08-05

**Authors:** Melissa J. Alldred, Harshitha Pidikiti, Kyrillos W. Ibrahim, Sang Han Lee, Adriana Heguy, Gabriela Chiosis, Elliott J. Mufson, Grace E. Stutzmann, Stephen D. Ginsberg

**Affiliations:** 1Center for Dementia Research, Nathan Kline Institute, Orangeburg, NY 10962, USA; melissa.alldred@nki.rfmh.org (M.J.A.); harshitha.pidikiti@nki.rfmh.org (H.P.); kyrillos.ibrahim@nki.rfmh.org (K.W.I.); sanghan.lee@nki.rfmh.org (S.H.L.); 2Department of Psychiatry, New York University Grossman School of Medicine, New York, NY 10016, USA; 3Genome Technology Center, New York University Grossman School of Medicine, New York, NY 10016, USA; adriana.heguy@nyulangone.org; 4Program in Chemical Biology, Sloan Kettering Institute, New York, NY 10065, USA; chiosisg@mskcc.org; 5Breast Cancer Medicine Service, Memorial Sloan Kettering Cancer Center, New York, NY 10065, USA; 6Department of Translational Neuroscience and Neurology, Barrow Neurological Institute, Phoenix, AZ 85013, USA; elliott.mufson@barrowneuro.org; 7Center for Neurodegenerative Disease and Therapeutics, The Chicago Medical School, Rosalind Franklin University, North Chicago, IL 60064, USA; grace.stutzmann@rosalindfranklin.edu; 8Department of Neuroscience & Physiology, New York University Grossman School of Medicine, New York, NY 10016, USA; 9NYU Neuroscience Institute, New York University Grossman School of Medicine, New York, NY 10016, USA

**Keywords:** Down syndrome, Alzheimer’s disease, trisomy, RNA-sequencing, laser capture microdissection, choline acetyltransferase, aging, maternal choline supplementation

## Abstract

Down syndrome (DS), stemming from the triplication of human chromosome 21, results in intellectual disability, with early mid-life onset of Alzheimer’s disease (AD) pathology. Early interventions to reduce cognitive impairments and neuropathology are lacking. One modality, maternal choline supplementation (MCS), has shown beneficial effects on behavior and gene expression in neurodevelopmental and neurodegenerative disorders, including trisomic mice. Loss of basal forebrain cholinergic neurons (BFCNs) and other DS/AD relevant hallmarks were observed in a well-established trisomic model (Ts65Dn, Ts). MCS attenuates these endophenotypes with beneficial behavioral effects in trisomic offspring. We postulate MCS ameliorates dysregulated cellular mechanisms within vulnerable BFCNs, with attenuation driven by novel gene expression. Here, choline acetyltransferase immunohistochemical labeling identified BFCNs in the medial septal/ventral diagonal band nuclei of the basal forebrain in Ts and normal disomic (2N) offspring at ~11 months of age from dams exposed to MCS or normal choline during the perinatal period. BFCNs (~500 per mouse) were microisolated and processed for RNA-sequencing. Bioinformatic assessment elucidated differentially expressed genes (DEGs) and pathway alterations in the context of genotype (Ts, 2N) and maternal diet (MCS, normal choline). MCS attenuated select dysregulated DEGs and relevant pathways in aged BFCNs. Trisomic MCS-responsive improvements included pathways such as cognitive impairment and nicotinamide adenine dinucleotide signaling, among others, indicative of increased behavioral and bioenergetic fitness. Although MCS does not eliminate the DS/AD phenotype, early choline delivery provides long-lasting benefits to aged trisomic BFCNs, indicating that MCS prolongs neuronal health in the context of DS/AD.

## 1. Introduction

Down syndrome (DS) is caused by the triplication of human chromosome 21 (HSA21) and is the most prevalent genetic cause of intellectual disability. Individuals with DS have increased long-term neurological disorders and most develop Alzheimer’s disease (AD) pathology during early mid-life [1,2], with concurrent cognitive decline resulting in early onset dementia [3,4]. DS in combination with AD dementia (DS + AD) occurs decades prior to sporadic AD, but recapitulates many of the pathological hallmarks, including amyloid-beta peptide (Aβ) containing senile plaques, tau-positive neurofibrillary tangles, cortical thinning, brain atrophy, and degeneration of basal forebrain cholinergic neurons (BFCNs) and the septohippocampal circuit [1,5,6,7,8,9]. Cholinergic basal forebrain projection neurons originate in the medial septum and project to the hippocampus, forming the bidirectional septohippocampal circuit [10,11]. BFCNs in the nucleus basalis of Meynert innervate the entire cortical mantle [12,13], making BFCN degeneration a critical subcortical hub of AD [14] and DS + AD [15] degeneration.

Since the Ts65Dn (Ts) mouse model of DS/AD recapitulates degeneration of the septohippocampal circuit [16,17,18], it has been used to study BFCN circuit dysfunction. In Ts65Dn mice, BFCN degeneration starts at ~6 months of age (MO), and progresses through ~10 MO [17,19,20], resulting in decreased cholinergic tone and aberrant activity of cholinergic projection neurons within the septohippocampal circuit [21,22].

Current DS and AD therapies are inadequate, with no treatments that slow or stop BFCN degeneration [23,24]. However, in the Ts65Dn trisomic model of DS/AD, maternal choline supplementation (MCS) has been shown to protect the phenotype of BFCNs in older animals [25]. MCS increases dietary choline to offspring during the perinatal period (e.g., gestation through weaning) [26,27,28]. MCS is based on the critical need for choline, an essential nutrient, during pregnancy and early life development. Human and rodent studies concur that current recommended daily choline intake are insufficient during pregnancy [29,30]. Choline plays three essential roles in the brain, serving as a precursor for the neurotransmitter acetylcholine, primary dietary methyl donor, and substrate for the phosphatidylethanolamine N-methyltransferase (PEMT) pathway. Acetylcholine is a key neurotransmitter essential during brain development for neuronal differentiation, proliferation, synaptic plasticity, and synapse formation [31,32,33]. The PEMT pathway is required for the generation of the neuronal membrane components phosphatidylcholine and sphingomyelin, among others [28,34,35]. Methylation is required for epigenetic regulation and programming of gene expression [33,36].

MCS provides beneficial effects to offspring in murine model systems of DS [37,38,39,40], Rett syndrome [41,42], and Fragile X [43], among other neurodevelopmental disorders. Children with DS were assayed for metabolite ratios by magnetic resonance imaging [44]. Significant decreases in choline/creatine ratios were seen in DS compared to controls in multiple brain areas, with lower metabolite ratios correlating with reductions in the intelligence quotient [44]. Human MCS clinical trials in normal pregnancy [30,45], and fetal alcohol syndrome [46,47,48], have shown improvements in learning and memory. In the Ts65Dn DS/AD mouse model, MCS improves learning, memory, and attentional behaviors [21,38,40,49,50], BFCN survival and activity [25,50], and attenuates the pathological enlargement of early endosomes in BFCNs [51]. MCS benefits are thought to be driven by epigenetic reprogramming of the PEMT pathway through alterations in the distribution and consumption of choline metabolites in the DS mouse model [34], which result in lifelong epigenetic reprogramming in the brain.

MCS provides beneficial effects on BFCN gene expression in ~6 MO [52] Ts65Dn mice at the onset of degeneration [53]. MCS attenuates basal forebrain cholinergic and parvalbumin neuronal loss at ~12 MO, increases septocholinergic activity to the hippocampus, and improves behavioral outcomes in multiple memory tasks in aged Ts65Dn mice [21,22,25,40]. However, it is critical to understand neuroprotective molecular and cellular alterations that maintain the septohippocampal connectome in response to MCS, which has important translational implications.

We interrogated BFCNs microisolated from ~11 MO animals, including Ts65Dn trisomic and normal disomic (2N) offspring from dams either fed a MCS or normal choline diet. Differentially expressed gene (DEG) analysis and downstream bioinformatic pathway inquiries were performed on BFCNs following single population RNA sequencing (RNA-seq), stratified by genotype and maternal diet as we previously reported [52]. We posit DEGs and biological pathways underlying the benefit of MCS for aged BFCNs will be elucidated.

## 2. Materials and Methods

### 2.1. Mouse Cohort Generation

Animal protocols were approved by the Nathan Kline Institute/New York University Grossman School of Medicine (NYUGSOM) IACUC in accordance with NIH guidelines. Breeder pairs (female Ts65Dn and male C57Bl/6J Eicher x C3H/HeSnJ F1 mice) were purchased from Jackson Laboratories (Bar Harbor, ME, USA) and mated at the Nathan Kline Institute. Breeder pairs were assigned to receive one of two choline-controlled experimental diets: (i) control rodent diet containing 1.1 g/kg choline chloride (AIN-76A; Dyets Inc., Bethlehem, PA, USA), or (ii) choline-supplemented diet containing 5.0 g/kg choline chloride (AIN-76A; Dyets Inc.), as described previously [22,54,55]. Offspring received choline normal or choline supplemented diet per their respective dams’ diet from embryonic day 0 (E0) to postnatal day 21 (P21) (Figure 1A). At weaning (P21) all offspring had ad libitum access to water and the control diet. The choline supplemented diet provides approximately 4.5 times the concentration of choline than the control diet but is still within the normal physiological range [56]. Tail clips were taken and genotyped [57] at weaning and mice were aged to ~10–12 MO (Figure 1A), with the average age of ~11 MO.

### 2.2. Tissue Preparation and Immunohistochemistry

Brain tissues were accessed from choline normal Ts65Dn (Ts; n = 11), MCS Ts65Dn (Ts+; n = 10), choline normal disomic (2N; n = 11) and MCS disomic (2N+; n = 12) male mice (age range: 10.43–11.93 MO, mean age 10.98 MO). Mice were transcardially perfused with ice-cold 0.15 M phosphate buffer [58,59]. Brains were removed from the calvarium and hemidissected using a bias cut ~1–1.5 mm lateral to the midline to preserve basal forebrain structures within the biased hemibrain [59,60], and immediately flash frozen and stored at −80 °C until sectioned for immunohistochemistry and laser capture microdissection (LCM). Cryostat sectioning (−25 °C) of 20 µm-thick tissue sections was performed in the coronal plane (CM1860UV, Leica, Buffalo Grove, IL, USA). A total of 3–6 sections per slide were mounted on polyethylene naphthalate (PEN) membrane slides. Slides were kept under desiccant at −80 °C prior to immunohistochemistry. PEN membrane slides were equilibrated to room temperature (RT) under desiccant (−20 °C for 5 min, 4 °C for 10 min, RT for 5 min) to prevent crystal formation, preserve tissue, and RNA integrity. A rapid staining protocol utilizing an antibody against choline acetyltransferase (ChAT) (AB144P, Millipore, Burlington, MA, USA) was performed to visualize cholinergic immunoreactive (ChAT-ir) BFCNs within the medial septal nucleus (MSN) while preserving intact RNA in the unfixed tissue as previously described [59,60]. RNase-free precautions were employed throughout the experimental paradigm. Solutions were made with 18.2 mega Ohm RNase-free water (Nanopure Diamond, Barnstead, Dubuque, IA, USA).

### 2.3. Laser Capture Microdissection

ChAT-ir BFCNs were identified within the MSN and isolated by LCM (LMD7000; Leica; Figure 1A) as previously described [59,60]. Approximately 500 ChAT-ir BFCNs were microisolated per brain before proceeding to RNA isolation and single population RNA-seq library preparation.

### 2.4. RNA Processing

A total of ~500 BFCNs were pooled per sample, and RNA was purified using the miRNeasy Micro kit (Qiagen, Germantown, MD, USA) according to manufacturers’ specifications. DNase digestion was performed twice sequentially before the final washes and RNA purification. RNA quality control (QC) was performed (RNA 6000 pico kit, Agilent, Santa Clara, CA, USA). Since most samples were below the limit of detection for RNA integrity (~50 pg/uL), RNA quality was not used to eliminate samples; rather, library preparation QC was used to determine viability for sequencing.

### 2.5. Library Preparation

The SMARTer Stranded Total RNA-Seq kit-Pico input Mammalian (v1, Takara Bio, Mountain View, CA, USA) was employed with minor modifications during step A to utilize the full volume of isolated RNA as previously described [60]. Samples were quantified (2100 HS DNA kit, Agilent), and those showing no peak between 300 and 500 base pairs were re-isolated by LCM, with RNA isolation and library preparation were repeated. Samples were isolated and processed in three independent batches, batch 1 (2N, n = 3; 2N+, n = 5; Ts, n = 4; Ts+, n = 5), batch 2 (2N, n = 2; 2N+, n = 2; Ts, n = 4; Ts+, n = 2), and batch 3 (2N, n = 6; 2N+, n = 5; Ts, n = 3; Ts+, n = 3). Cohorts were processed for bioinformatics: 2N, n = 11; 2N+, n = 12; Ts, n = 11; Ts+, n = 10. Power calculations performed utilizing samples from batch 1 and ~6 MO MSN data [60] indicated 10+ samples per cohort were needed for the aged population (~11 MO) to achieve 80% power for statistical significance at *p* = 0.05.

### 2.6. Single Population RNA-Seq

Library samples were pooled in equimolar concentrations and assayed on an HiSeq-4000 (Illumina, San Diego, CA, USA) using a single-read 50-cycle protocol at NYUGSOM Genome Technology Center (GTC). Raw FastQ files were assessed for quality control using FastQC (v0.11.9) [61]. Trimmomatic (v0.39) was used to trim any adapter sequences [62]. A decontamination step was performed at the read level to remove any contaminants, using a customized Kraken2 (v2.1.13) [63] database, limiting retained reads to the mouse reference genome (GRCm39). This step improved data integrity, alignment accuracy, and quantification. Reads were indexed and aligned to GRCm39 using the STAR aligner (v2.7.10a) [64]. Gene-level quantification was performed using RSEM (v1.3.3) to obtain a raw count matrix for differential gene expression analysis. Post-alignment and RNA-seq-specific quality metrics were collected using Picard (v2.27.1) [65].

### 2.7. Statistical Analysis

Differential gene expression analysis was performed on the gene-level count matrix obtained using RSEM. Genes were retained if they scored over 0.1 counts per million (CPM) in more than 50% of the samples to filter lowly expressed genes, decrease statistical errors, and false discovery rates [66,67]. To reduce batch processing technical variability, multiple batch correction methods were tested and applied to increase robustness of the differential expression results. Combat-seq (CS) corrects batch effects before modeling the data using empirical Bayes adjustment on the raw counts data, while estimating and removing batch-related biases [68,69]. Random effects (RE) modeling directly incorporates each batch as a random factor within the linear mixed model (LMM), where it accounts for batch-to-batch variability while retaining the overall structure of the data without the loss of biological variation [70]. In the Variance Partition (VP) approach, gene expression values are adjusted based on the estimation of variation due to the batch effects and simultaneously removing this variation, which results in the corrected gene expression values used for the differential expression analysis [71].

In all three approaches, trimmed mean of M-values (TMM) normalization was applied to calculate normalization factors to correct the differences in library size and composition, which are introduced during linear modeling to ensure accurate comparisons. Differential gene expression was performed using the DREAM pipeline [72] built on top of limma-voom from the variancePartition package [71]. In each of the three workflows, additional covariates such as intergenic rate and usable base percentage were included to account for variation, along with a biological variable called Group representing the combination of genotype and maternal diet. Except for Group, the other covariates were derived from RNA-seq reads using Picard. DEGs were identified for all the three approaches separately using the topTable function, with significance defined as a *p*-value < 0.05 (Appendix A). To ensure robustness and consistency, significant genes common to at least two of the three batch correction approaches were considered DEGs. For visualizations and pathway analyses, numerical data and statistics generated from the CS method were used for DEGs. DEGs significant in VP and RE (and not CS) utilized VP numerical data and statistics and were annotated as ‘VP’ (Appendix A). The remaining genes were considered non-significant, regardless of *p*-value.

### 2.8. Bioinformatic Pathway Analyses

Pathway analyses consisted of Ingenuity Pathway Analysis (IPA; Qiagen) [73,74] and Gene Ontology (GO) [75,76]. Shiny package (v.1.8.1.1) was utilized to create a web-based app to run GO analysis using R version 4.4.0/RStudio v1 + 494. This app was also used to filter keyword targets to identify classes of processes affected by genotype and region [52,77,78,79]. “Rescue” by MCS was defined if the following criteria were met: (i) significant dysregulation was seen in Ts vs. 2N, (ii) significant alteration in expression was seen in Ts+ vs. Ts, and (iii) activation scores showed increased activity in Ts vs. 2N and decreased activity in Ts+ vs. Ts (or vice versa). GO analysis could not replicate activation and samples were therefore classified as “MCS responsive” using the first two criteria above. Venn diagrams were made using VennDiagram v1.7.3 in R and InteractiVenn [80]. Heatmaps were generated in Graphpad Prism (v. 10.4.1; Boston, MA, USA). Gene network plots were generated via igraph v2.1.4 using R v4.4.2.

## 3. Results

In aged DS mice, MCS rescues degeneration of BFCNs [25] and improves septohippocampal dependent behavioral outcomes [28,38,49]. To determine underlying molecular phenotypes modulated by MCS, we isolated ChAT-ir BFCNs via LCM and performed single population RNA-seq in the context of genotype and maternal diet at ~11 MO (Figure 1A) for DEG and bioinformatic inquiry (Figure 1B, Appendix A). Multidimensional scale plots (MDS; CS shown) indicated that the inter- and intra-group variability were high for ~11 MO BFCN gene expression (Figure 1C). Normalized cell counts confirmed the specific isolation of excitatory neurons using cell-type-specific gene markers (*p* < 1.84 × 10^−5^ or lower; Appendix A) [81]. In the normal choline diet paradigm, mice were compared by **genotype** (e.g., Ts vs. 2N). Trisomic mice in the context of maternal diet were compared by **disease diet** (e.g., Ts+ vs. Ts). Additional comparisons included a **disomic diet** comparison (e.g., 2N+ vs. 2N), a **supplemented genotype** comparison (e.g., Ts+ vs. 2N+), and a **diet plus genotype** comparison (e.g., Ts+ vs. 2N; Appendix A, respectively) as previously described [52].

DEGs were operationally defined as genes significantly different (*p* < 0.05) in two or more batch effects programs by genotype (Figure 1D, top) or disease diet (Figure 1D, bottom). Volcano plots show significant DEGs using log_2_ fold change (LFC) taken from the CS analysis for those DEGs present in all three comparisons, and/or VP and CS and/or RE and CS by genotype (Figure 2A, Appendix A) and disease diet (Figure 2B; Appendix A). For DEGs identified by VP and RE, VP was used to generate LFC values, with gray dots representing those DEGs that were not significant (ns) in at least two batch effects comparisons. Bar charts are utilized to visualize the DEG distribution in 0.10 LFC increments. DEGs in the genotype comparison showed more downregulated than upregulated DEGs (Figure 2C), while the disease diet comparison revealed a more even distribution, although there were still more downregulated DEGs compared to upregulated DEGs (Figure 2D). The majority of DEGs displayed LFCs >2, likely due to the batch effect normalization. Comparing genotype to disease diet DEGs, 113 DEGs were MCS responsive, with 110 DEGs being “rescued by MCS”, indicating they are downregulated by genotype and upregulated by disease diet (or upregulated by genotype and downregulated by the disease diet), accounting for 21.1% of the genotype and 15.1% of the disease diet DEGs (Figure 2E).

Triplicated gene expression, independent of significance level, was upregulated in the genotype comparison, with little effect in the disease diet comparison (Figure 3A). A total of eight HSA21 ortholog DEGs and five non-orthologous triplicated DEGs were significantly upregulated by genotype (Figure 3B, left). A total of three DEGs were rescued by MCS, with one DEG additive from MCS (Figure 3B, middle). Moreover, disease diet had two unique upregulated and two unique downregulated DEGs for HSA21 orthologs with one non-orthologous triplicated gene downregulated (Figure 3B, right). These results indicate MCS has modest effects on triplicated DEGs, similar to previous findings in ~6 MO BFCNs [52].

To interrogate potential MCS-responsive mechanisms in aged MSN BFCNs, IPA and GO analyses were performed using DEGs (*p* < 0.05) from the genotype and disease diet comparisons. IPA canonical pathways, as well as disease and functions (D/Fs) were examined in the genotype and disease diet comparisons, and cross-referenced between analyses to determine pathways rescued by MCS and pathways uniquely modulated by genotype or disease diet. Overall, genotype differences showed greater dysregulation, including more total dysregulated pathways (Figure 4A) and higher activation (z-score; Figure 4B) than disease diet via IPA canonical pathway analysis. Several pathways were uniquely dysregulated by genotype, including downregulation of autophagy, complex I biosynthesis, glutaminergic and GABAergic receptor signaling, NGF signaling and signaling by NTRK1 (TrkA; Figure 4C; Appendix A). Unique to the disease diet comparison were downregulation of pathways including oxidative stress induced senescence, pyruvate metabolism, and RNA polymerase II transcription, and upregulation and SUMOlyation of DNA replication proteins, and necroptosis signaling pathways (Figure 4D; Appendix A). MCS responsive pathways in disease diet showed a rescue effect compared to genotype (by z-score) with many associated with recycling and normalization of cellular signaling functions. This included upregulation of macroautophagy, mitochondrial protein degradation, DNA signaling and protein sorting signaling, along with downregulation of neurexins and neuroligins, reelin signaling, and sirtuin signaling (Figure 4E). A few pathways displayed additive effects with genotype and MCS compounding activation status (Figure 4E, bottom), which suggest that some dysregulated mechanisms are compensatory to BFCN survival or function, such as downregulating the CLEAR signaling pathway (Figure 4E). The top select pathways rescued by MCS were analyzed, and several DEGs rescued by MCS were common to multiple canonical pathways (Figure 4F). To determine whether MCS-rescued pathways were unique to older ChAT-ir BFCNs, pathways from the current ~11 MO cohort were compared to previously published ~ 6 MO MSN BFCNs [52]. Aged neurons displayed differential activity regulation compared to ~6 MO BFCNs in pathways including opioid and reelin signaling (Appendix A). However, some pathways were rescued by MCS at both timepoints including the sirtuin signaling pathway, while others were rescued by MCS at ~6 MO but were no longer significantly rescued in ~11 MO BFCNs, including the key pathways oxidative phosphorylation and CLEAR signaling (Appendix A).

D/F analysis examines how dysregulated genes and pathways affect neurological and cellular functions. Alterations observed by genotype were more severe than by disease diet for total number of D/Fs (Figure 5A) and for activation status, as deternimed by z-score (Figure 5B). Alterations in genotype suggest ~11 MO trisomic BFCNs are highly dysfunctional, indicated by D/Fs “degeneration of cells”, “accumulation of vesicles”, “senescence of cells “and “stress response of cells”, which were all significantly upregulated without significant attenuation by MCS (Figure 5C; Appendix A). In the disease diet comparison, MCS uniquely modulated upregulation of metabolism of membrane lipid derivative, and sensitivity of cells, among others, while turnover of lipids was downregulated (Figure 5D; Appendix A). MCS at this older age point still exhibited phenotypic rescue in multiple D/Fs, including apoptosis, cognitive impairment, and phosphorylation of proteins (Figure 5E). Like IPA canonical pathways, D/Fs showed additive effects including quantity of neurons and transcription (Figure 5E, bottom). MCS-responsive DEGs were further analyzed for driver DEGs in several rescued D/Fs (Figure 5F). Select driver DEGs mirrored canonical pathway drivers, including microtubule-associated protein 1 light chain 3 alpha (*Map1lc3a*) and heat-shock protein 9 (*Hspa9*) (Figure 4F and Figure 5F). However, some driver DEGs were unique for canonical pathways, including mitogen-activated protein kinase kinase 4 (*Map2k4*), basic helix–loop–helix ARNT-like 1 (*Bmal1*), and glutamate dehydrogenase 1 (*Glud1*) (Figure 4F) or D/Fs, including activin A receptor, type 1B (*Acvr1b*), catalase (*Cat*), and intersectin 1 (*Itsn1*) (Figure 5F). Consistent with canonical pathways, a few D/Fs were rescued by MCS at ~6 MO and ~11 MO, notably apoptosis. Many D/Fs were either rescued in older animals, including phosphorylation of protein and receptor mediated endocytosis, or were additive in both ~6 MO and ~11 MO, including quantity of neurons, transcription, and expression of RNA (Appendix A). Modulation by MCS was also apparent when analyzing diet plus genotype (Ts+ vs. 2N; Appendix A) by IPA analysis, with the majority of selected pathways and D/Fs exhibiting rescue or attenuation of genotypic dysregulation.

GO analysis was performed to examine overall effects of genotype and disease diet DEGs from ~11 MO BFCNs. The resulting biological processes, cellular components and molecular functions were binned into 15 biologically relevant categories using Shiny app, with the non-neuronal processes eliminated from analysis. Genotype-specific changes were principally cellular, with activity and protein processes being the most abundant categories (Figure 6A; Appendix A). Compared to the disease diet paradigm, behavioral, RNA and synaptic processes showed a higher overall percentage of significant processes relative to genotype, with actin and signaling showing relatively fewer processes (Figure 6A,B; Appendix A). The most abundant categories in disease diet were activity, protein, and cellular processes (Figure 6B; Appendix A). There were significant processes unique to either genotype or disease diet, with processes significant in both considered to be MCS responsive. GO analysis of each category determined the most MCS responsive processes were in axonal/dendritic (~41% overlap), behavioral (~45% overlap), cellular (~35% overlap), and protein (~38% overlap) categories (Figure 6C; Appendix A).

## 4. Discussion

Single population RNA-seq was performed to determine the impact of MCS in ~11 MO MSN BFCNs in a Ts65Dn trisomic murine model of DS/AD. Bioinformatic inquiry determined underlying cellular and molecular alterations at a timepoint when MSN cholinergic neuron degeneration is fully realized. A genotype-dependent dysregulation of 522 DEGs was found in trisomic BFCNs within the medial septum, which were linked to dysregulated pathways and processes that drive neurodegeneration. These neurodegenerative DEGs and pathways were assessed in offspring following early choline supplementation. At ~11 MO, trisomic BFCNs displayed 729 disease diet-dependent DEGs, with 110 DEGs rescued by MCS, i.e., dysregulated in the genotype comparison and attenuated in the disease diet comparison. Rescued genes drive cellular mechanisms linked to learning and memory retention, indicating perinatal choline delivery results in beneficial epigenetic reprogramming of key genes and pathways in behaviors that decline in individuals with DS as they age.

Single population RNA-seq revealed an upregulated signature of HSA21 orthologs in BFCNs. A few (3) upregulated HSA21 orthologs were attenuated by MCS (Figure 3B), indicating MCS does not eliminate the genetic DS signature in aged trisomic BFCNs, commensurate with findings reported from previous trisomic mouse [78,79,82] and postmortem human DS neurons [77,83,84]. In aged BFCNs, triplicated HSA21 orthologs account for a small percentage of the total differential gene signature. This suggests MCS has therapeutically relevant beneficial effects without solely targeting triplicated transcripts. Additional mechanisms and driver DEGs need to be identified to confirm MCS therapeutic efficacy, although the present study is highly supportive in a vulnerable cell type within the degenerated septohippocampal circuit.

MCS had more profound effects on trisomic BFCNs than either genotype alterations or the impact of MCS in disomic animals. This included a greater quantity of DEGs and more uniquely regulated genes in trisomic mice, indicating MCS has a stronger impact when a neurodegenerative phenotype is present. Beneficial MCS alterations in the context of neurodegeneration support previous findings indicating a correlation between greater dysregulation and greater MCS responsiveness, either due to aging [85,86,87] and/or trisomy [25,51,60]. Although MCS did not fully restore genotype level dysregulation in trisomic ~11 MO trisomic BFCNs, rescue accounted for ~15.1% of the disease diet DEGs (Figure 2E).

MCS enabled rescue of multiple canonical pathways and D/Fs, notably signaling pathways. Nicotinamide adenine dinucleotide (NAD) signaling pathway is downregulated by genotype and upregulated by disease diet. This is a critically important cellular signaling pathway, with downregulation previously linked to AD progression [88]. NAD+ precursors increase cell viability and reduce beta-site amyloid precursor protein cleaving enzyme 1 (*Bace1*) and amyloid precursor protein (*App*) metabolites in a dose dependent manner [89]. Notably, MCS uniquely downregulated oxidative stress-induced senescence in trisomic BFCNs (Figure 4D). Increased oxidative stress and accumulation of reactive oxygen species have been linked to AD, Parkinson’s disease, and cardiovascular disease [90]. Although mitochondrial dysfunction was increased in ~11 MO BFCNs, MCS rescued mitochondrial protein degradation pathways (Figure 4E). The rescue effect of MCS was restricted to select oxidative phosphorylation genotype DEGs, including NADH–ubiquinone oxidoreductase subunit B1 (*Ndufb1*) and succinate dehydrogenase complex, subunit C, an integral membrane protein (*Sdhc*) (Appendix A). This likely indicates that MCS attenuates, but does not stop genotype-driven bioenergetic dysregulation, although ~6 MO BFCNs do show significant amelioration [52], suggesting an age-related decline in MCS effectiveness.

Autophagy and endocytosis were downregulated in ~11 MO BFCNs (Figure 5C) without being rescued by MCS. However, upregulation in apoptosis was rescued by MCS, supporting previous work demonstrating MCS increased survival of ChAT-ir MSN BFCNs in trisomic mice [25]. D/F analysis revealed 27 DEGs were rescued by MCS, including the driver DEG, *Map1lc3a* (Figure 5F; Appendix A), which has been implicated in autophagy during AD progression [91,92]. Additional DEGs were rescued in the apoptosis pathway and several other D/Fs, including genotypic downregulation of *Cat*, an antioxidant enzyme that reduces reactive oxygen species which are impaired in AD [93]. *Itsn1*, triplicated in DS and overexpression inhibits endocytosis [94], was significantly downregulated by MCS (Figure 3B) and linked to multiple rescued D/Fs (Figure 5F). *Hspa9* is downregulated in AD, causing mitochondrial fragmentation and dysfunction [95]. In trisomic neurons, *Hspa9* is a driver gene of the mitochondrial protein degradation pathway (Figure 4F), as well as multiple D/Fs (Figure 5F), indicating rescuing *Hspa9* is critical for multiple cellular functions in aged BFCNs. These DEGs are highlighted as regulators of multiple mechanisms dysregulated in aged DS MSN BFCNs, suggesting targets for potential interventional therapeutics.

GO analysis classified processes linked to disease and MCS responsiveness by cellular function. Overlaps between pathways dysregulated by genotype and modulated by disease diet were found and termed “MCS responsive”, as GO does not delineate activation status of significant processes. “Behavioral” processes were identified as highly MCS responsive, along with “axonal–dendritic” processes, although the total number of dysregulated and MCS responsive processes in these categories was relatively low compared to the total number of dysregulated neuronal processes. “Protein” and “cell” processes were highly MCS responsive and accounted for 15.2% and 34.7% of the overall disease diet processes affected. At this age, cellular function and internal protein modifications are most MCS responsive, suggesting early choline delivery affects neurodegenerative mechanisms responsible for normative cellular function and survival, which was confirmed by comparing findings from younger ~6 MO [52] with older ~11 MO BFCNs. We found “axonal–dendritic” and “protein” processes were MCS responsive throughout the adult lifespan, although “neurotransmitter” and “synaptic” processes showed higher relative MCS responsiveness in younger animals [52]. “Behavioral” and “cell” processes were not tabulated in the younger cohort and could not be compared. Overall, GO analysis confirmed IPA analysis, showing lifelong benefits of MCS in vulnerable MSN BFCNs.

A caveat of this study is the lack of female mice. Previous studies revealed mixed results with human and mouse mixed-sex studies showing no significant gene expression alterations in DS [58,84,96], while a sex-dependent effect of cholinergic activity has been observed [97]. An age-matched female cohort is currently ready to examine gene expression stratified by sex in BFCNs. Moreover, effect sizes and large LFCs of significant genes indicate there are likely genes that show small or variable expression alterations eliminated during the batch effect normalization approach. A less stringent but permissive bioinformatic method, such as weighted gene co-expression analysis (WGCNA), could be employed on these data. However, previous analysis on a younger BFCN (~6 MO) cohort did not reveal many functional alterations unique to WGCNA compared to differential gene expression analysis [60]. Future profiling of BFCNs localized to the nucleus basalis of Meynert, which display a tau pathology in prodromal and early AD [98,99,100,101], are essential to examine basocortical circuit alterations in response to MCS. Profiling will also include basal forebrain inhibitory neurons, which are MCS responsive in the trisomic model [25]. Single population RNA-seq of these basal forebrain populations is critical, as unique neuronal signatures have been seen in trisomic single population studies of excitatory hippocampal neurons [78,79]. Alternative animal models of DS/AD could be considered, such as TcMac21 [102], which encodes a triplicated copy of the entire human HSA21 chromosome in mice to confirm validity of targets and to identify potential novel targets not present in this initial analysis. Future studies may also include follow-up of driver genes identified herein for translational impact as a therapeutic intervention in iPSCs derived from individuals with DS as well as AD for mechanistic studies [103].

## 5. Conclusions

Single population RNA-seq of trisomic MSN BFCNs displayed significant gene expression dysregulation in ~11 MO trisomic mice compared to disomic counterparts. These underlie the degenerative endophenotype seen in aged trisomic mice. MCS partially attenuates these dysregulated genes, with ~21% of the genotype dependent DEGs rescued by early choline delivery. MCS also modulates a significant subset of unique genes in ~11 MO trisomic BFCNs. Genes and pathways modulated by MCS show a significant impact of this early dietary intervention. Further, these preclinical data directly identify cellular machinery impacted by early choline delivery, with long-term effects on pups suggesting choline insufficiency can have lifelong detrimental consequences in adult offspring. Aggregating these findings demonstrate MCS provides significant functional amelioration of vulnerable cellular pathways, biological functions and disease phenotypes associated with trisomy that also informs AD. Key DEGs and pathways have been defined as high priority candidates for future mechanistic and translational research including mitochondrial fitness/bioenergetics, cell viability, neurotrophic support, and synaptic integrity. Interestingly, many of these target DEGs and pathways discovered in the context of DS are also implicated in the onset of AD, suggesting these driver genes directly affect neurodegeneration of BFCNs, a hallmark of DS and AD pathobiology. Future validation of MCS is essential for potential intervention in disease progression in DS and AD, as in vivo preclinical models demonstrate greater choline demands in DS compared to normative development. Preservation of BFCN circuity is essential to maintain memory and executive function during the onset and progression of dementia, and MCS may be one viable intervention that attains this goal with minimal side effects or toxicity.

## Figures and Tables

**Figure 1 biomolecules-15-01131-f001:**
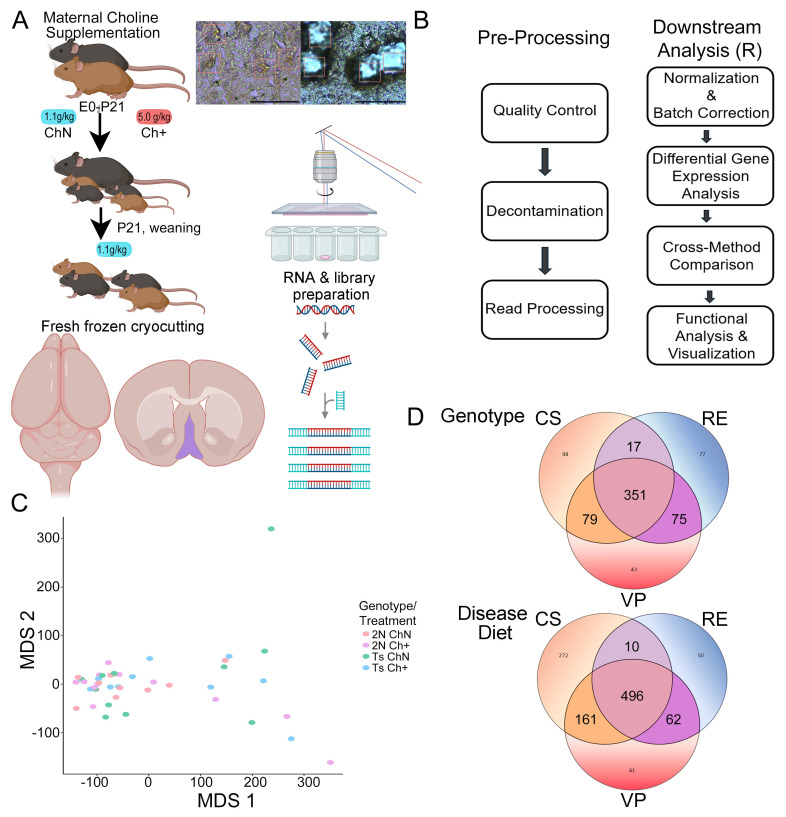
(**A**) Overview of the MCS paradigm and LCM collection of ~11 MO MSN BFCNs for RNA-sequencing, with ChAT-ir neurons shown. Scale bar: 50 µm. (**B**) Schematic representation of bioinformatic processing, including decontamination and batch effects normalization procedures. (**C**) MDS plots show variable expression of BFCNs from the 2N (salmon, n = 11), 2N+ (purple, n = 12), Ts (green, n = 11), and Ts+ (blue, n = 10) cohorts, with each dot representing an individual animal. (**D**) Venn diagrams show Combat seq (CS), Random effects (RE), and Variance Partition (VP) DEGs for genotype comparison (Ts vs. 2N, **top**) and disease diet comparison (Ts+ vs. Ts, **bottom**). Bolded DEGs significant in two or more batch correction methods were employed for downstream analysis.

**Figure 2 biomolecules-15-01131-f002:**
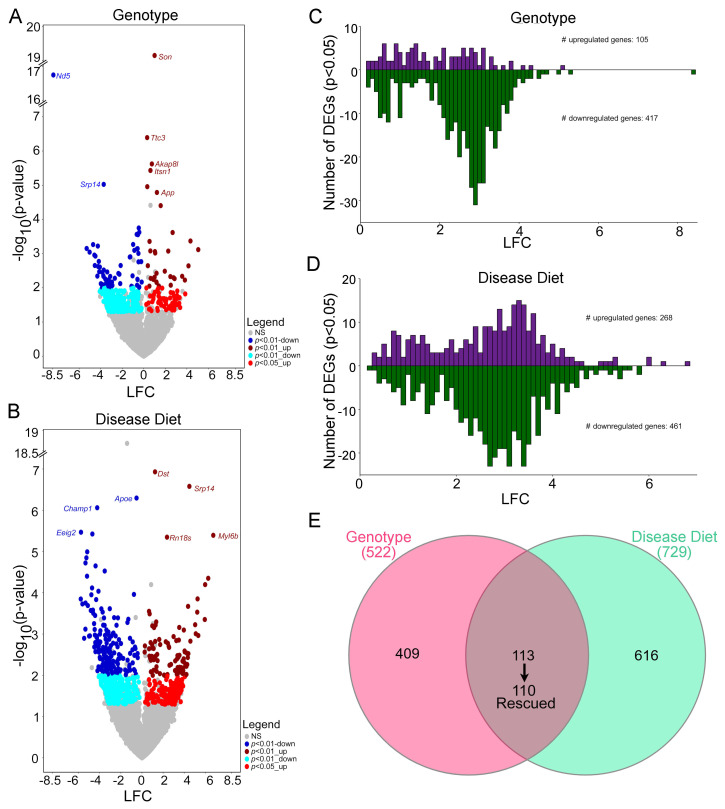
DEGs are compared for regulation status and significance. (**A**) Volcano plots demonstrate upregulation and downregulation of genotype (Ts vs. 2N) DEGs. (**B**) Disease diet volcano plots display upregulation and downregulation of MCS-responsive genes. Key: gray = not significant (ns, as described in the Materials and Methods and Figure 1); dark blue = *p* < 0.01 downregulation; light blue = *p* < 0.05 downregulation; dark red = *p* < 0.01 upregulation; red = *p* < 0.05 upregulation. (**C**,**D**) Bar charts indicate on the *y*-axis the number of DEGs binned by 0.25 log fold change (LFC, *x*-axis) indicting more downregulated than upregulated DEGs in genotype (Ts vs. 2N; **C**) and disease diet (Ts+ vs. Ts; **D**). (**E**) Venn diagram shows overlap of DEGs from genotype and disease diet comparisons.

**Figure 3 biomolecules-15-01131-f003:**
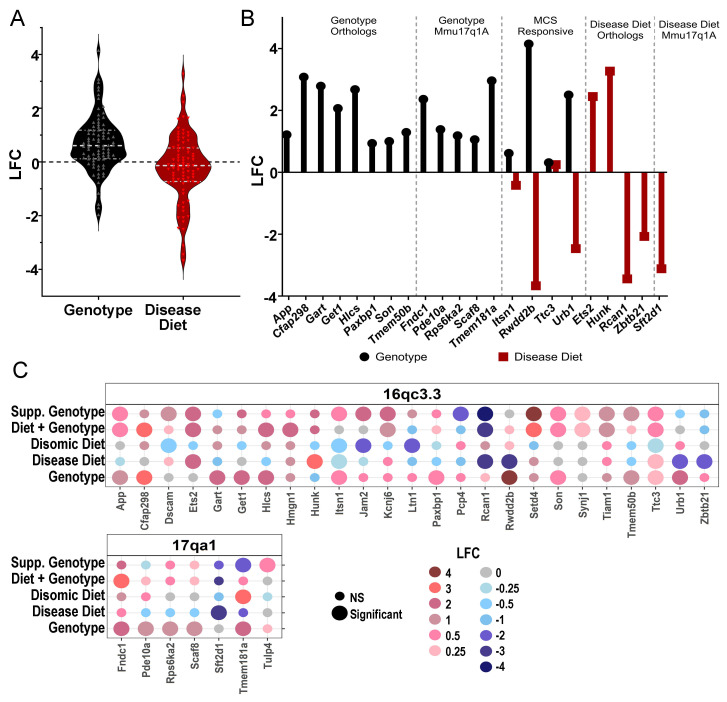
Trisomic genes in the MSN revealed a mild response to MCS. (**A**) Total expressed genes by genotype (black) and disease diet (red) comparisons indicate mean LFC (*y*-axis) is upregulated by genotype, but MCS has little effect (LFC mean near zero, dashed line). (**B**) DEGs upregulated by genotype (black) were from triplicated HSA21 orthologs as well as from Mmu17q1a. MCS rescued a few DEGs. Select triplicated DEGs were also uniquely modulated in the disease diet comparison (red). (**C**) Heatmap indicates DEGs significantly affected in one or more dietary comparison. Key: significance represented by a large dot, not significant with a small dot. LFC ranged on a scale of dark blue (highly downregulated) to dark red (highly upregulated).

**Figure 4 biomolecules-15-01131-f004:**
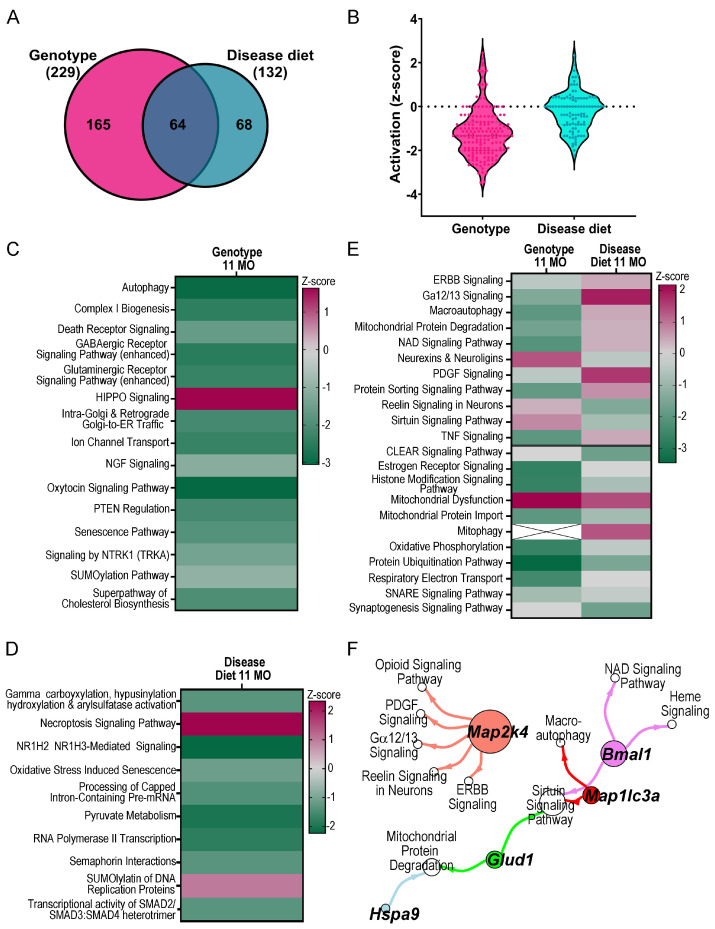
(**A**) Total number of dysregulated pathways in genotype and disease diet are shown by Venn diagram. (**B**) Activation of each pathway is represented by dots within violin plots of all neuronal canonical pathways for genotype (pink) and disease diet (teal). (**C**) Canonical pathways differentially regulated by genotype indicated the majority of pathways were downregulated with relatively few upregulated pathways. Select pathways show ~11 MO trisomic BFCNs have significant dysregulation in normal metabolic and cellular functions. (**D**) MCS had unique effects on trisomic MSN BFCNs at ~11 MO, with select canonical pathways shown by heatmap. (**E**) Genotype dependent pathways were altered by MCS, displaying rescue of select canonical pathways (**top**) and additive effects in a subset (**bottom**), with a gray line differentiating rescued and additive pathways. (**F**) Driver DEGs are highlighted from MCS rescued canonical pathways as identified in (**E**). Key: z-score: magenta = upregulation; green = downregulation; x indicates IPA could not assess activation status.

**Figure 5 biomolecules-15-01131-f005:**
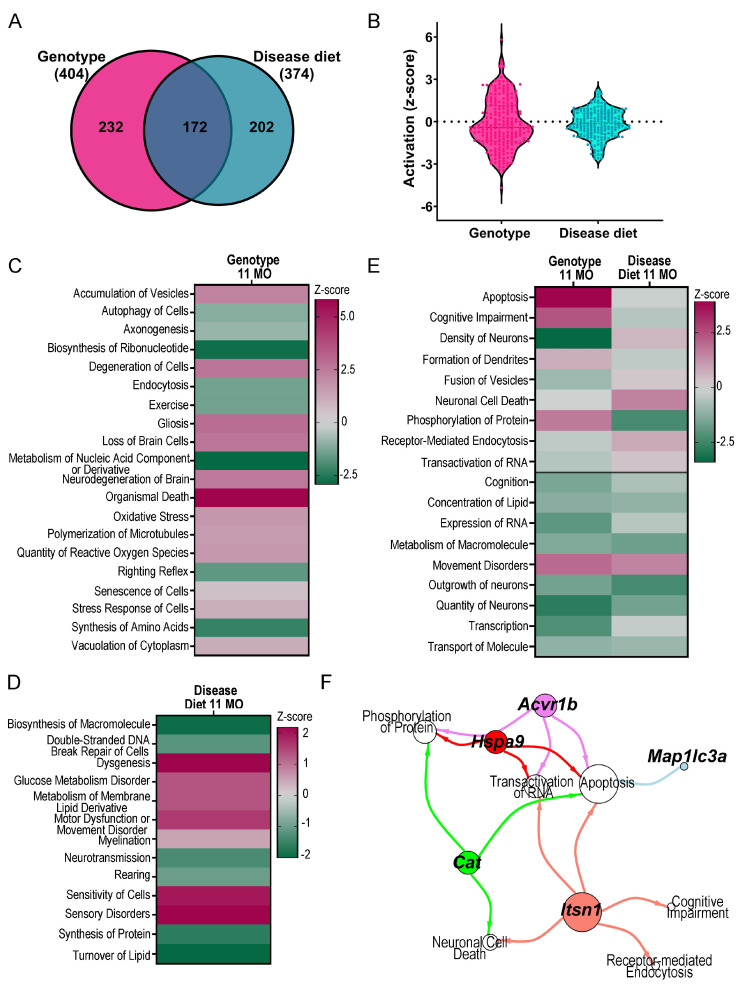
(**A**) Total number of dysregulated D/Fs in genotype and disease diet are shown by Venn diagram. (**B**) Activation of each D/F are represented by dots within violin plots of all neuronal canonical pathways for genotype (pink) and disease diet (teal). (**C**) D/Fs displayed unique dysregulation by genotype. The majority of D/Fs shown were upregulated with a few D/Fs downregulated. (**D**) Heatmap indicates unique effects of MCS on trisomic BFCNs at ~11 MO. (**E**) Genotype dependent D/Fs were rescued by MCS as seen by disease diet z-score reversals (**top**), while select D/Fs showed additive effects of dysfunction in trisomic BFCNs (**bottom**). Gray line differentiating rescue and additive D/Fs. (**F**) Driver DEGs are highlighted from the rescued D/Fs as identified in (**E**) Key: z-score: magenta = upregulation; green = downregulation; x indicates IPA could not assess activation status.

**Figure 6 biomolecules-15-01131-f006:**
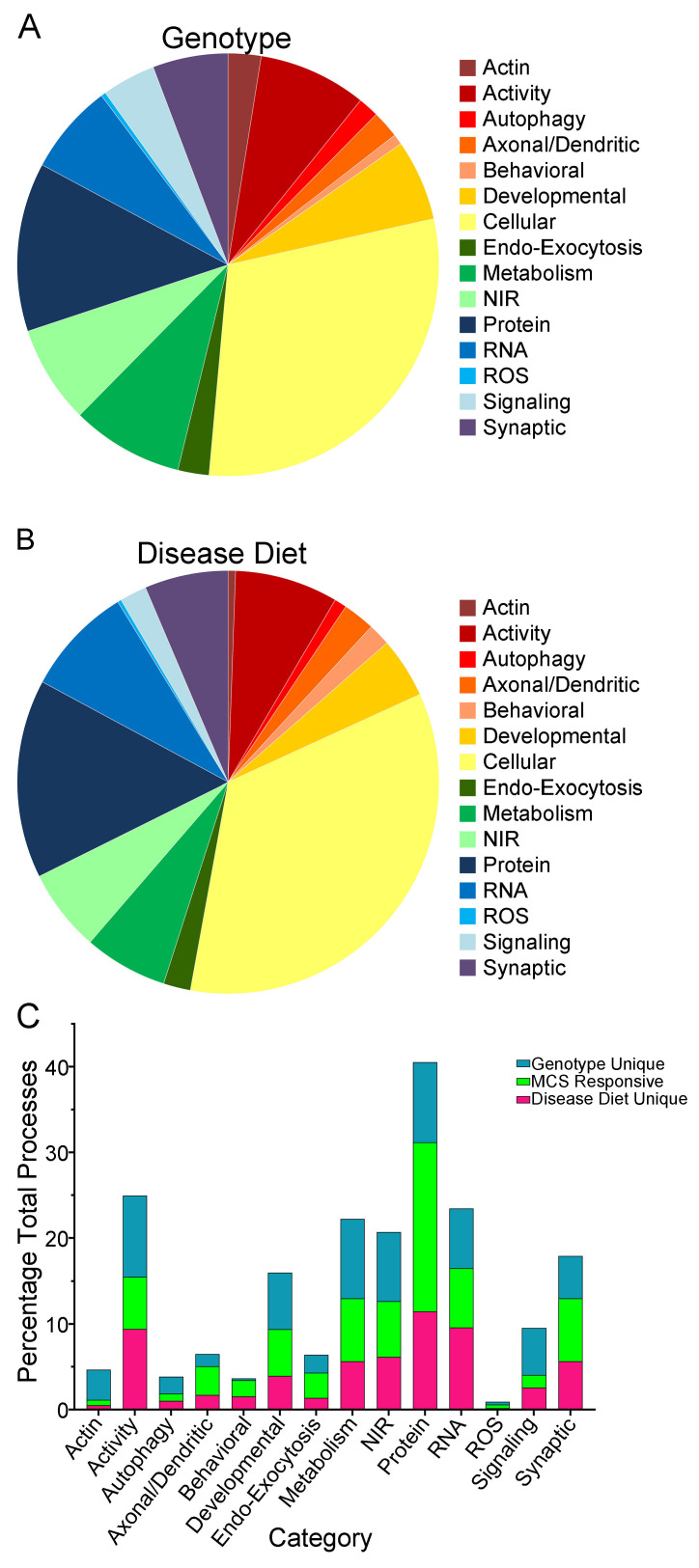
GO analysis demonstrated select genotype alterations were MCS responsive in specific categories, with other categories showing little MCS responsivity. (**A**) Genotype effects were binned to 15 categories with a pie chart showing relative abundance of processes altered in ~11 MO BFCNs. (**B**) Disease diet processes were binned to the same categories but showed a unique mosaic of processes affected by MCS. (**C**) Comparing genotype to disease diet, MCS responsive (green) processes were affected by both genotype and disease diet. Genotype (teal) and disease diet (magenta) specific processes were also identified in each category. Number of processes is indicated on the *y*-axis and categories are defined on the *x*-axis.

## Data Availability

The datasets supporting the conclusions of this article are available as follows. RNA-seq data analyzed within this study are available from GEO (www.ncbi.nlm.nih.gov/geo; GSE296095; accessed on 30 April 2025) or from the corresponding author upon request. LCM images for brain tissue and RNA/library metadata are available from the authors upon request or from synapse.org SYN66724706.

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
