# Peer review of "Benefits of Maternal Choline Supplementation on Aged Basal Forebrain Cholinergic Neurons (BFCNs) in a Mouse Model of Down Syndrome and Alzheimer’s Disease"

_biomolecules, 2025, doi:10.3390/biom15081131_

Round 1
Reviewer 1 Report
Comments and Suggestions for Authors
Reviewer comments and suggestions
The present research paper has analyzed the current landscape of the Down syndrome (DS), stemming from triplication of human chromosome 21, results in intellectual disability, with early mid-life onset of Alzheimer’s disease (AD) pathology. Maternal choline supplementation (MCS) attenuates these endophenotypes with beneficial behavioral effects in trisomic offspring. The study postulates MCS ameliorates dysregulated cellular mechanisms within vulnerable BFCNs, with attenuation driven by novel gene expression. Here, choline acetyltransferase immunohistochemical labeling identified BFCNs within in the medial septal/ventral diagonal band nuclei of the basal forebrain in Ts and normal disomic (2N) offspring at ~11 months of age in offspring from dams exposed to MCS or normal choline during the perinatal period. BFCNs (~500 per mouse) were micro isolated and processed for RNA-sequencing. Bioinformatic assessment elucidated differentially expressed genes (DEGs) and pathway alterations in the context of genotype (Ts, 2N) and maternal diet (MCS, normal choline). The results showed that MCS attenuated select dysregulated DEGs and relevant pathways in aged BFCNs. Trisomic MCS-responsive improvements included pathways such as cognitive impairment and nicotinamide adenine dinucleotide signaling, among others, indicative of increased behavioral and bioenergetic fitness.
Decision: Minor revision is needed.
The paper has nicely compiled and covered the utmost part related to maternal choline supplementation as a promising intervention to improve neuronal health and cognitive outcomes in Down syndrome and Alzheimer’s disease models. However, in many places, the authors need to do a few corrections in the manuscript. Based on my view, below are some comments that need to be incorporated in the revised version of the manuscript.
- Line 30: “within in the” should be “with the”.
- The authors have added too many references at several places that are not required please add the appropriate references and describe it well
- Line 41: “long lasting” should be hyphenated: “long-lasting”. Please make sure that this is corrected throughout the paper.
- Lines 102-103: The phrasing of this sentence is awkward. This is an example of a less repetitive way to phrase this sentence: “We posit that gene expression changes and pathways underlying the benefit of MCS for aged BFCNs will be elucidated.”
- Please mention the date of ethical approval for the study as I could not see in the manuscript
- Line 279: “cross referenced” should be hyphenated: “cross-referenced”.
- Line 434: The phrasing of “...more total DEGs and more genes uniquely regulated...” is slightly awkward. It can be fixed by restating it in the following way: “...a greater quantity of total DEGs and more uniquely regulated genes...”
- Line 494: The lack of female mice is mentioned, but this paper would benefit from describing this limitation further in the discussion and conclusion sections since it is significant to the study.
- Please check Figure 4 to see whether that is present in the manuscript or not.
- The conclusion paragraph could be made better if it briefly included the impacts and significance of this study. Speaking about broader implications for these clinical trials and the next steps in therapy development would strengthen this paragraph.
- All references should be modified based on the MDPI journal guidelines.
Author Response
Reviewer #1
We thank the Reviewer for their thorough review and insightful comments. We emended the manuscript accordingly in a point-by-point fashion below.
- “Line 30: “within in the” should be “with the.”
Response: Corrected.
- “The authors have added too many references at several places that are not required please add the appropriate references and describe it well.”
Response: Thank you for this comment. We removed several references and clarified existing references to reflect presented data as follows: line 64, line 72, line 78, lines 96-97, line 115, line 127, line 165, line 229, and line 437.
- “Line 41: “long lasting” should be hyphenated: “long-lasting”. Please make sure that this is corrected throughout the paper.”
Response: Corrected.
- “Lines 102-103: The phrasing of this sentence is awkward.”
Response: Based on this inquiry, we emended the text (lines 105-106).
- “Please mention the date of ethical approval for the study as I could not see in the manuscript.”
Response: Thank you for catching this oversight. We revised the information in the section titled “Institutional Review Board Statement” (lines 566-568) accordingly.
- “Line 279: “cross referenced” should be hyphenated: “cross-referenced”.
Response: Corrected.
- “Line 434: The phrasing of “...more total DEGs and more genes uniquely regulated...” is slightly awkward.”
Response: Based on this inquiry, we emended the text (line 434).
- “Line 494: The lack of female mice is mentioned, but this paper would benefit from describing this limitation further in the discussion and conclusion sections since it is significant to the study.”
Response: We thank the Reviewer for this suggestion and have emended the caveats section to include more detail as to the limitations of the present report (lines 498-501).
- “Please check Figure 4 to see whether that is present in the manuscript or not.”
Response: We are not sure why the Reviewer could not see Figure 4. Based on comments raised by Reviewer #2 (below), Figure 4 has been revised and we ensure that it is present in the resubmission.
- “The conclusion paragraph could be made better if it briefly included the impacts and significance of this study. Speaking about broader implications for these clinical trials and the next steps in therapy development would strengthen this paragraph.”
Response: We thank the Reviewer for this thoughtful suggestion. We revised the conclusions paragraph (lines 526-530 and 538-540) to include this vital information.
- “All references should be modified based on the MDPI journal guidelines.”
Response: Thank you for this inquiry. We formatted the references per journal guidelines for the resubmission.
Reviewer 2 Report
Comments and Suggestions for Authors
On the whole, the paper appears to be sound. The authors did a good job outlining their reasoning and presenting the data. Some questions remain. Are choline levels known to be particularly low in DS patients? Do low choline levels in non-DS patients affect cognition in later life? Did the authors measure adult choline levels in pups exposed to MCS? IF the authors do not expect a long term impact on choline levels, how do they expect MCS to impact pathways in 11 MO mice? I may have missed it, but is it known if MCS eases Abeta or Tau burden in DS-AD?
While the authors do mention it as a caveat, this study uses only male mice. Why? Is there some difference between the sexes in the trisomy models used?
I would expect that, as they near weaning,, the pups do start to ingest some of the chow present in the cage. Do the authors think that it's likely that direct supplementation at that age has no effect?
Note that I think most of this can be resolved either in a rebuttal or with modifications to the text, though the figure issue needs to be corrected.
It would be nice to see the comparison between the disomic (2N) and treated trisomic (Ts+) groups (referred to as Diet + genotype in the paper) in more of the figures in order to compare the scale of impact of genotype versus treatment on the designated pathways. The authors do point out in the text that genotype seems to have a far larger impact than the MCS on the investigated pathways and included some of this information in the supplemental figures, but figures 4 and 5, in particular, are lacking this visual representation of that data.
Author Response
Reviewer #2
We thank the Reviewer for their helpful comments regarding this submission. We appreciate the queries and respond in a point-by-point fashion below.
- “Are choline levels known to be particularly low in DS patients? Do low choline levels in non-DS patients affect cognition in later life?”
Response: We revised text in the Introduction to include the known deficits in choline levels in DS (lines 82-85), Longitudinal studies in individuals with DS or non-DS counterparts have not been performed, so the effect early low choline has on adults later in life in terms of cognitive performance is currently understudied and remains to be addressed.
- “Did the authors measure adult choline levels in pups exposed to MCS? IF the authors do not expect a long term impact on choline levels, how do they expect MCS to impact pathways in 11 MO mice?”
Response: Thank you for this comment. Unfortunately, measurement of choline levels in adult offspring was not performed. Studies using MCS have shown epigenetic reprogramming as indicated in the Introduction (lines 91-92).
- “I may have missed it, but is it known if MCS eases Abeta or Tau burden in DS-AD?”
Response: Thank you for this important question. The short response is that to date, no one has measured potential benefits in terms of pathological hallmarks in individuals with DS-AD. Murine DS mouse models do not develop amyloid plaques or neurofibrillary tangles. For future reference, research in mouse models of cerebral amyloid overexpression have demonstrated that MCS does attenuate amyloid plaque burden (e.g., Judd JM, et al., Inflammation and the pathological progression of Alzheimer’s disease are associated with low circulating choline levels. Acta Neuropathol. 2023 Oct;146(4):565-583. PMCID: PMC10499952; Mellott TJ, et al., Perinatal choline supplementation reduces amyloidosis and increases choline acetyltransferase expression in the hippocampus of the APPswePS1dE9 Alzheimer’s disease model mice. PLoS One. 2017 Jan 19;12(1):e0170450. PMCID: PMC5245895).
- “While the authors do mention it as a caveat, this study uses only male mice. Why? Is there some difference between the sexes in the trisomy models used?”
Response: Thank you for this inquiry. We expanded the caveats section accordingly on lines 498-501. Please also see Reviewer #1, comment 8.
- “I would expect that, as they near weaning, the pups do start to ingest some of the chow present in the cage. Do the authors think that it's likely that direct supplementation at that age has no effect?”
Response: This is an interesting question. Direct supplementation is possible near the end of weaning just prior to switching offspring to the normal choline diet (1.1g/kg) for the rest of the lifespan. We acknowledge individual pups may consume choline supplemented chow, but we are unable to quantify intake. We ensure conformity by limiting MCS from embryonic day 0 (E0) to postnatal day 21 (P21) (lines 115-118), where the overwhelming source of choline intake is through the dam.
- “It would be nice to see the comparison between the disomic (2N) and treated trisomic (Ts+) groups (referred to as Diet + genotype int he paper) in more of the figures in order to compare the scale of impact of genotype versus treatment on the designated pathways.”
Response: In response to this comment, we added text to the Results (lines 352-356) and added additional panels to Supplementary Figure 4 to compare these pathways and D/Fs to the diet plus genotype. A full analysis of this additional subgroup is beyond the scope of the present work.
- “The authors do point out in the text that genotype seems to have a far larger impact than the MCS on the investigated pathways and included some of this information in the supplemental figures, but figures 4 and 5, in particular, are lacking this visual representation of that data.
Response: To address this inquiry, we added text to the Results (lines 281-283 and lines 330-332), 2 panels to Figure 4 (now Fig. 4A-F) and 2 panels to Figure 5 (now Fig. 5A-F), and the respective Figure Legends (lines 295-297 and lines 358-360).